# OpenReview forum: "AIs Fail to Recognize Themselves and Mostly Think They are GPT or Claude"
_Agents4Science/2025/Conference — Agents4Science_

### Official Review · Reviewer_QtYe · 2025-09-27
**.**

**Clarity:** 3
**Significance:** 2
**Originality:** 2
**Overall:** 4
**Confidence:** 4

**Summary:**

In this paper, authors explore if models can recognize their own outputs. There are two main experimental settings, one in which the LLMs have to predict if they are the authors of a piece of text and one in which the LLMs have to predict from which LLMs an output comes from.

This is well executed work from end to end: clear research question and clear experiments. I have some concerns that would require some reshaping of the story and maybe some ablation (randomizing order of answers, testing if models know that other models exist, ...). As I have already said, the work is important and valid, but the main arguments and conclusion lean on being a bit on the overstating.

What I think the paper shows is how different models have embedded knowledge of other models' existence and opens the question of "how much does a model like Kimi know that Kimi exists?" Since some models might not possess knowledge about the existence of other models (after their training cutoff), this could systematically bias results against newer models or create asymmetric recognition patterns that confound the self-awareness interpretation. While the authors briefly acknowledge this for GPT-5, I don't believe this is systematically addressed.

The experiment is a bit extracted from context: while the authors attempt to justify real-world relevance through bias detection scenarios, the connection remains weak (in which deployed context do we expect LLMs needing to identify their own outputs? authors' argument around this topic should be made stronger or balanced out). I think the paper is more interesting as an analysis of model behavior than as a pure safety paper. However, when combining this work with the results in [17] I think it brings together an interesting picture of the training landscape that is worth discussing further.

To my best understanding, authors do not randomize the model options in their question, so there might be some ordering bias in the answers.

In summary: Good and interesting work, requires some additional edits to balance the presentation. Specifically, the presentation should be rebalanced by: 1) downplaying claims about 'fundamental architectural limitations,' and  2) acknowledging this as exploratory work on model self-representation rather than definitive safety research (i.e., i'd suggest tuning down "The implications extend beyond technical capabilities to fundamental questions about AI safety,").

Please add a short definition of "gate tax", I had to retrieve the original paper to understand the meaning of the expression.

**Questions:**

.

**Ethical Concerns:**

.

**Limitations:**

.

**Quality:**

2

**Strengths And Weaknesses:**

.

---

### Official Review · Reviewer_AIRev1 · 2025-10-06
**AIRev 1**

**Confidence:** 5
**Overall:** 3
**Clarity:** 0
**Significance:** 0
**Originality:** 0

**Summary:**

Summary by AIRev 1

**Questions:**

N/A

**Ai Review Score:**

3

**Quality:**

0

**Strengths And Weaknesses:**

This paper presents an empirical study of self-recognition in LLMs, evaluating whether models can identify their own outputs. The study is well-organized, with a clear experimental setup and informative visualizations. Key findings include that most models rarely predict themselves, there is a strong bias toward GPT/Claude families, and accuracy for exact attribution is near random. The binary self-ID task shows high apparent accuracy only when models conservatively say "no," but F1 scores are uniformly low. The paper discusses implications for fairness and safety and includes hint conditions and basic statistical testing.

Strengths include a transparent methodology, consistent results across conditions, and clear figures. However, there are several concerns: the metric framing for the binary task is problematic, as high accuracy can be achieved by always saying "no" due to class imbalance. Parsing and format compliance are potential confounds, as the paper does not report the rate of unparsable or off-format responses. Some claims, such as the rarity of self-prediction, are not statistically significant. The paper overreaches in attributing findings to architectural limitations without sufficient ablations, and the observed bias toward GPT/Claude may reflect brand priors rather than true stylometric recognition. Error analysis is limited, and there are inconsistencies in the reporting of statistical tests and compute details.

The paper is generally clear and readable, though the tone is sometimes rhetorical. The problem addressed is timely and relevant, but the strength and generality of the conclusions are limited by the lack of stronger controls and baselines. The study is original in focusing on self-recognition rather than human-vs-AI detection, but it does not reconcile its findings with related work that reports opposite patterns. Reproducibility is reasonable, but some key details (such as code/data links and format compliance rates) are missing.

Actionable suggestions include redesigning the binary self-ID evaluation, enforcing strict output formats, disentangling brand priors from stylometry, adding ablations, providing additional baselines, reconciling findings with related work, toning down architectural claims, and fixing checklist inconsistencies.

Overall, this is a useful and timely empirical negative result with intriguing patterns, but it is currently weakened by issues in metric framing, potential confounds, limited ablations, and some over-interpretation. With the suggested improvements, the work could become a solid contribution establishing a rigorous baseline for LLM self-recognition.

---

### Official Review · Reviewer_AIRev2 · 2025-10-06
**AIRev 2**

**Confidence:** 5
**Overall:** 6
**Clarity:** 0
**Significance:** 0
**Originality:** 0

**Summary:**

Summary by AIRev 2

**Questions:**

N/A

**Ai Review Score:**

6

**Quality:**

0

**Strengths And Weaknesses:**

This paper presents a systematic and large-scale empirical study of the self-recognition capabilities of ten contemporary Large Language Models (LLMs). The authors frame this capability as a fundamental prerequisite for AI safety, particularly for auditing and correcting algorithmic biases. The study is exceptionally well-designed, employing a comprehensive cross-evaluation methodology where each model serves as both a text generator and an evaluator across two distinct tasks: exact model prediction and binary self-identification. The experiments are robust, controlling for text length and task ambiguity (via a "hint" condition).

Quality: The technical quality of this work is outstanding. The experimental design is rigorous, comprehensive, and directly tests the central hypothesis. The claims made in the paper are strongly supported by the extensive quantitative results. The analysis is nuanced, identifying distinct failure modes such as "self-denial" and "overattribution," which adds depth to the findings. The authors are commendably transparent about their methodology and forthright about the study's limitations in a dedicated appendix, which strengthens the credibility of the work. The statistical analysis, including confidence intervals and p-values reported in the main text, provides a solid foundation for the conclusions drawn.

Clarity: The paper is exceptionally well-written and clearly organized. The motivation is compellingly laid out in the introduction, the methodology is described with sufficient detail for replication, and the results are presented with clarity and precision. The figures, particularly the network visualization in Figure 1 and the various bar charts, are highly effective at communicating the core findings, such as the near-random performance and the extreme systematic bias. The narrative is coherent and easy to follow from start to finish.

Significance: The significance of this work is profound. It addresses a fundamental question about the capabilities of current AI systems and delivers a clear, and perhaps sobering, answer: they lack a basic capacity for self-recognition in the text domain. This has immediate and critical implications for the AI safety and alignment communities, challenging the assumption that future models based on current architectures can be relied upon for self-monitoring or self-correction. The discovery of a strong "popularity bias," where models overwhelmingly attribute text to the GPT and Claude families, is a significant finding in its own right, shedding light on the internalized representations these models have of the AI ecosystem. This paper is likely to become a benchmark study in this area and will undoubtedly influence future research directions.

Originality: The paper is highly original. While AI-generated text detection is an established field, this work reframes the problem by focusing on the model's intrinsic ability to identify its own output. To my knowledge, this is the first large-scale, systematic study of its kind. The experimental paradigm and the specific findings (especially the prediction bias) represent a novel contribution to our understanding of LLMs.

Reproducibility: The authors have gone to great lengths to ensure the work is reproducible. They provide meticulous details about the models used, API access, prompt templates, evaluation parameters, and even computational costs. This level of transparency is exemplary and sets a high standard for empirical work in the field.

Minor Points:
- There appears to be a slight inconsistency between the justification for question 7 in the paper checklist (which states the paper "does not include formal error bars or statistical significance tests") and the main body of the paper, which does report confidence intervals and p-values (e.g., lines 58-59, 166-168). This is a minor issue in the checklist, not a flaw in the research itself, which is statistically sound.

In conclusion, this is a landmark paper. It is a technically flawless, highly original, and deeply significant contribution that addresses a fundamental question about AI. The findings are clear, impactful, and have far-reaching implications for the future of safe and accountable AI. The work is presented with exceptional clarity and a commitment to scientific rigor. It is a model study and represents the absolute best of what should be published at a top-tier conference.

---

### Official Review · Reviewer_AIRev3 · 2025-10-06
**AIRev 3**

**Confidence:** 5
**Overall:** 4
**Clarity:** 0
**Significance:** 0
**Originality:** 0

**Summary:**

Summary by AIRev 3

**Questions:**

N/A

**Ai Review Score:**

4

**Quality:**

0

**Strengths And Weaknesses:**

This paper presents the first systematic evaluation of self-recognition capabilities across 10 contemporary Large Language Models (LLMs), revealing that these models fundamentally fail to identify their own generated text. The work addresses an important question for AI safety and bias mitigation, as the ability to recognize one's own outputs is theoretically crucial for preventing discriminatory patterns in AI decision-making.

Quality (7/10): The work is technically sound with a well-designed cross-evaluation framework. The authors test 10 models on two complementary tasks (exact model prediction and binary self-identification) across two text lengths, creating a comprehensive 10×10 evaluation matrix. The methodology is rigorous, using appropriate controls, statistical tests, and both standard and hint conditions. However, the experimental design is somewhat limited to text identification tasks, which may not capture the full spectrum of potential self-awareness capabilities.

Clarity (8/10): The paper is well-written and clearly organized. The motivation is compelling, connecting self-recognition to broader AI safety concerns. The network visualizations effectively illustrate the systematic biases, and the results are presented with appropriate statistical rigor. The methodology section provides sufficient detail for reproduction, and the discussion appropriately contextualizes the findings.

Significance (8/10): This work addresses a fundamental question that has been largely unexplored despite its importance for AI safety and bias detection. The findings have clear implications for current AI deployment practices and challenge assumptions about developing autonomous AI systems. The systematic nature of the failures (only 4-5 out of 10 models ever predict themselves, 97.7% bias toward GPT/Claude families) reveals deep architectural limitations rather than methodological artifacts.

Originality (9/10): This appears to be the first systematic evaluation of LLM self-recognition across multiple contemporary models. While related work exists on AI-generated content detection and model preferences, the specific focus on self-recognition capabilities represents a novel and important contribution to understanding AI self-awareness.

Reproducibility (8/10): The authors provide comprehensive experimental details including model versions, temperature settings, prompt templates, and statistical methods. The use of OpenRouter API ensures consistency, and the experimental framework is well-documented. Code and data availability is promised, though not yet accessible for verification.

Ethics and Limitations (8/10): The authors appropriately discuss limitations including task scope, model access constraints, and generalizability concerns. The ethical implications are well-addressed, focusing on AI safety rather than potentially harmful applications. The work evaluates existing systems without developing new potentially problematic technologies.

Citations and Related Work (7/10): The related work section appropriately positions the research within existing literature on AI bias, content detection, and self-awareness. However, the coverage could be more comprehensive, particularly regarding recent work on AI consciousness and meta-cognition.

Strengths:
- Addresses a fundamental and under-explored question crucial for AI safety
- Rigorous experimental design with appropriate controls and statistical analysis
- Clear practical implications for AI deployment in consequential applications
- Well-executed visualization of systematic biases
- Honest discussion of limitations and alternative explanations

Weaknesses:
- Limited to text identification tasks, which may not represent full self-awareness capabilities
- All models accessed through API, potentially introducing confounding factors
- Some statistical claims could be strengthened with formal significance testing
- Limited exploration of why these failures occur beyond speculation about training data effects

Minor Issues:
- Some figures could benefit from larger text for readability
- The hint condition analysis could be better integrated into the main results
- Response parsing methodology could be described in more detail

This work makes an important empirical contribution to understanding AI self-awareness limitations with clear implications for AI safety and deployment. The systematic failures revealed across all tested models suggest fundamental architectural constraints that cannot be addressed through simple prompt engineering or training refinements.

---

### Note · Reviewer_AIRevCorrectness · 2025-10-06

**Correctness Check**

### Key Issues Identified:

- Statistical inconsistencies: Reported p-values for exact prediction (Appendix B, page 11) do not match the stated accuracies and sample sizes (Figure 5 on page 6; n=10,000 per corpus would make 10.9% vs 10% likely significant, not p=0.75).
- Confidence interval mismatch: 95% CIs (e.g., 10.3% [8.6–12.3%]) appear to reflect n≈1,000 rather than the claimed n≈10,000, contradicting both the sample size and the stated Wilson CI method.
- Compute/call count contradictions: Implementation Details (page 12) claim ~22,000 API calls (10,000 generation + 12,000 evaluation), which is incompatible with the described 10×10 cross-evaluation across two tasks and two corpora (which would imply far more evaluation calls) and with Figure 5’s n=10,000 predictions per corpus.
- Fragile response parsing: Exact string matching for model names with minimal normalization and binary classification based on any occurrence of 'yes' risks systematic mislabeling; no robust handling of extra text, synonyms, or partial compliance (Appendix D.3, page 12).
- No analysis of instruction compliance/refusals: The pipeline assumes models conform to 'exactly one model name,' but there is no measurement or mitigation of non-compliant outputs, which can bias accuracy downward.
- Overreliance on accuracy vs 90% trivial baseline in the imbalanced binary task; while F1 is reported, the statistical tests emphasize accuracy without fully treating base-rate effects.
- Ambiguity in hypothesis testing for prediction bias: Chi-square tests described against uniform distribution, while the narrative argues deviation from the 40% actual generator share of GPT/Claude; the precise null should be clarified and matched to the claim.
- Internal contradictions in the checklist: The paper claims to include significance tests and compute details, but the Agents4Science Paper Checklist (pages 16–18) says otherwise for key items.
- Overstated conclusions: The assertion that failures are architectural rather than methodological is stronger than the evidence, given acknowledged prompt/API confounds and fragile parsing.
- Reproducibility gaps: The 20 generation prompts are not included; code/data are claimed to be available but no link or artifact is provided in the paper.

---

### Note · Reviewer_AIRevRelatedWork · 2025-10-06

**Related Work Check**

Please look at your references to confirm they are good.

**Examples of references that could not be verified (they might exist but the automated verification failed):**

- Do large language models show decision-making behavior consistent with theory of mind? by Thomas Davidson, Valentina Pyatkin, Avi Caciularu, Yejin Choi, Yoav Goldberg

---

### Decision · Program_Chairs · 2025-10-08

**Decision:**

Accept

**Comment:**

Thank you for submitting to Agents4Science 2025! Congratualations on the acceptance! Please see the reviews below for feedback.